# OpenReview forum: "STAC: When Innocent Tools Form Dangerous Chains to Jailbreak LLM Agents"
_ICLR.cc/2026/Conference — ICLR 2026 Conference Withdrawn Submission_

### Official Review · Reviewer_6MEV · 2025-10-26

**Soundness:** 2
**Presentation:** 3
**Contribution:** 2
**Rating:** 2
**Confidence:** 4

**Summary:**

The paper introduces STAC, a multi-turn red-teaming framework that targets tool-enabled LLM agents. Its central contribution is the composition of individually benign-looking tool calls whose cumulative effect becomes harmful only at the final step, i.e., each intermediate action appears innocuous but together they enable a covert attack. While the concept is novel and compelling, the manuscript currently lacks concrete empirical evidence or real-world incident analysis demonstrating that such staged, cumulative attacks are feasible or observed in practice.

**Strengths:**

- Introduces STAC as a distinct, sequence‑level vulnerability for tool‑enabled agents and operationalizes it via a closed‑loop generation–verification–prompting pipeline
- Comparisons with single‑turn and adapted multi‑turn baselines show large margins

**Weaknesses:**

- The STAC pipeline assumes near-white-box access to precise tool schemas, parameters, and environment state to construct and verify attack chains, an assumption that is often unrealistic in real-world agent deployments.
- The paper lacks a formal threat model and does not specify attacker capabilities, goals, or assumptions, undermining the rigor expected of AI-security research. Explicitly state attacker knowledge (tool schemas, environment state) given that the Generator/Verifier are granted such access; evaluate STAC when tool metadata is partially hidden or noisy.
- The entire STAC pipeline heavily depends on large LLMs for generation, verification, planning, and judging, yet the paper does not justify why alternative non-LLM or hybrid approaches (e.g., symbolic reasoning, static verification, or rule-based execution tracing) would be infeasible. Moreover, the authors provide no discussion of how hallucinations or false positives from these LLM components are mitigated, which undermines the methodological rigor and reproducibility of the proposed framework.
- The Judge is GPT‑4.1, the same model family used for generation/verification/planning. This risks correlated failure modes and optimistic ASR/PH/RR estimates.
- Experiments are in SHADE‑Arena/ASB sandboxes. Add results on a public agent stack (e.g., OS/desktop/web workflows) with realistic tools and rate limits to assess generalization.
- Only prompt‑based defenses are tested; discuss or prototype action‑level guards (e.g., precondition checkers, state‑diff risk analysis, sequence‑level monitors).

**Questions:**

- Threat Model Clarification: What are the assumed attacker capabilities and knowledge? Does STAC require white-box access to tool schemas, parameters, and environment state? Please evaluate whether the attack still succeeds when tool metadata is partially hidden or perturbed.

- LLM Dependence and Hallucination Control: Given that all components (Generator, Verifier, Planner, Judge) rely on GPT-4.1, how do you ensure consistency, reduce hallucinations or false positives, and verify that results are not artifacts of model bias?

- Model Independence of Evaluation: Since the Judge shares the same model family as other components, did you test cross-model or human-verified evaluation to confirm ASR/PH/RR robustness?

- Practical Deployment: How practical is STAC under realistic conditions where agents operate with limited tool visibility, API rate limits, or partially observable states (e.g., OS-level or web-based agents)?

---

> ### Author Response · Authors · 2025-12-03
> **Response to Reviewer 6MEV (Part 1)**
>
> We thank the reviewer for their constructive feedback. We address each weakness and question raised below.
>
> ---
>
> ### Weakness 1
>
> We acknowledge that the STAC pipeline assumes attacker access to tool schemas, parameters, and environment information. This assumption is appropriate for our current evaluation setting, where experiments are conducted in **controlled Python-based sandbox environments** designed specifically to expose tool interfaces. Our goal in this paper is to demonstrate the existence and feasibility of multi-turn, state-dependent tool-attack chaining under clear and well-defined conditions.
>
> In real-world deployments, full white-box access may indeed be limited. However, attackers often possess **partial or indirect visibility** into available tools, data, and environment state—for example, through documentation, error messages, observable effects of tool calls, API contracts, or agent outputs. Such visibility is already sufficient to enable tool-chain orchestration. We therefore view STAC as modeling a conservative and analytically transparent baseline rather than an unrealistic threat.
>
> Evaluating how STAC’s effectiveness changes under **weaker information assumptions** (e.g., partial tool schemas, black-box probing, noisy environment feedback) is an important next step. A systematic study of these factors is beyond the scope of the present paper, but our benchmark provides a useful starting point for future work in this direction.
>
> Additionally, we’d like to highlight that the vulnerabilities demonstrated in STAC have already been exploited in real-world deployments. Anthropic’s recent public case study on AI-enabled espionage (https://www.anthropic.com/news/disrupting-AI-espionage) highlights adversaries who **“broke down their attacks into small, seemingly innocent tasks that Claude would execute without being provided the full context of their malicious purpose”** in an agentic tool-use setting. This demonstrates that multi-step manipulation based on partial information is a realistic threat model.
>
> ### Weakness 2
>
> We appreciate the reviewer’s emphasis on formalizing the threat model. As noted in our response to Reviewer aDEf’s Weakness 1, our threat model assumes an **adversary who controls the user-facing channel** of an agentic system and can directly issue prompts. This adversary has access to **tool schemas and environment information** to the extent that such information is exposed to (or inferable by) any user of the system. Our use of white-box access in the Generator/Verifier is addressed in our response to your Weakness 1.
>
> We agree that clarifying these assumptions explicitly will improve rigor, and we will strengthen the paper by formalizing the attacker’s capabilities, goals, and access level.
>
> Evaluating STAC under **partial, noisy, or incomplete tool metadata** is an important extension—but beyond the scope of this paper. This would require systematically varying information access conditions and measuring attack degradation, which we identify as valuable future work. Our goal in the present work is to establish that multi-step, state-dependent harm is feasible even under idealized conditions; future work can explore robustness under more restrictive threat models.
>
> ### Weakness 3
>
> We appreciate the reviewer raising this point. We do **not** claim that non-LLM or hybrid approaches are infeasible; exploring symbolic reasoning, static verification, or rule-based tracing is an interesting direction for future work. Our choice to use LLMs throughout the STAC pipeline is driven by **practicality and generalizability**, not exclusivity.
>
> STAC is designed to model attacks on **realistic agentic systems**, which often operate in complex, partially specified, or dynamically changing environments. Even though our evaluation environments are controlled Python sandboxes, real-world deployments involve:
>
> - large, heterogeneous toolsets,
> - high-dimensional or unstructured state spaces,
> - ambiguous or user-defined tool semantics, and
> - interactions that cannot be fully enumerated symbolically.
>
> Large LLMs are uniquely suited for these conditions because they can flexibly reason, plan, and adapt across diverse tools and tasks. In practice, we also found that **smaller models (e.g., Qwen3-32B)** were insufficient for key pipeline roles such as chain generation and planning due to weaker reasoning ability—hence the choice of GPT-4.1 for those components.
>
> Regarding hallucinations and false positives, we mitigate their impact in two ways:
>
> 1. **Verification through environment execution:**
>    Any tool call proposed by the Generator or Verifier is actually executed in the environment. If a hallucinated or incorrect call is proposed, the environment immediately rejects it, and the Verifier revises the chain—preventing hallucinations from propagating.

---

> > ### Author Response · Authors · 2025-12-03
> > **Response to Reviewer 6MEV (Part 2)**
> >
> > 2. **Manual spot checks of the Judge model:**
> >    We manually inspected a subset of judged trajectories and found high agreement with human assessments of harmfulness, providing confidence in reliability.
> >
> > ### Weakness 4
> >
> > We acknowledge the concern that using GPT-4.1 as the Judge may introduce correlated biases with the Generator. This design choice is **intentional** and serves two practical purposes:
> >
> > **1. Ensuring consistent evaluation criteria.**
> > The Judge must assess whether a tool call meaningfully advances the malicious goal and whether a prompt appears harmful. Using the same model family ensures that the attacker and evaluator operate with a **shared notion of harmfulness**, reducing variance caused purely by cross-model disagreement. Heterogeneous Judges can introduce noise when models differ in how they interpret tool semantics or evaluate intent.
> >
> > **2. Verifying sequences through environment execution rather than model belief.**
> > Crucially, the Judge does **not** determine success based solely on model reasoning. Success is tied to **actual environment state changes** and **agent outputs**, which are externally validated by executing tool calls in the sandbox. This provides a ground-truth signal that mitigates the risk of optimistic scoring driven by model-model agreement.
> >
> > We recognize that exploring heterogeneous Judges (e.g., different model families or external rule-based safety checkers) would yield additional insight and may reveal cross-model divergence. Such analysis is an important direction for future work, but our goal in the present paper is to provide a **clean, controlled evaluation** of the STAC pipeline rather than to study Judge diversity.
> >
> > ### Weakness 5
> >
> > We agree that assessing STAC in real-world agent stacks (e.g., OS, desktop, or web automation systems) would provide valuable evidence of external validity. However, this falls outside the scope of the present work, which is focused on the **scientific characterization of the vulnerability class** rather than deployment-level testing.
> >
> > SHADE-Arena and Agent-SafetyBench were chosen because they provide **controlled, reproducible, and well-specified environments** that allow us to isolate and analyze multi-turn, state-dependent tool-use vulnerabilities without confounding factors that stem from real-world deployments. These simulator-based environments are standard in academic research.
> >
> > We consider evaluating STAC on production-grade agent stacks an important direction for **future applied work**, but such experiments require additional engineering resources and system access beyond what is feasible in a single research paper. Our contribution lays the groundwork for these follow-up evaluations.
> >
> > ### Weakness 6
> >
> > We agree that action-level defenses represent important directions for strengthening agentic safety. However, the primary focus of this paper is on **developing and systematizing the attack side**: constructing multi-turn, state-dependent tool-attack chains and demonstrating their feasibility across diverse environments. The prompt-based defenses we evaluate are intended only as **baseline references**, chosen because they are lightweight, broadly applicable across environments, and agnostic to models, environments, and tasks.
> >
> > We fully agree that exploring action-level defenses and testing STAC under richer guardrail mechanisms is a valuable direction for future research. Our STAC benchmark provides a foundation upon which these more sophisticated, action-level safety systems can be systematically evaluated.
> >
> > ---
> >
> > ### Question 1
> >
> > Please see our responses to Weaknesses 1 and 2.
> >
> > ### Question 2
> >
> > Please see our responses to Weaknesses 3 and 4.
> >
> > ### Question 3
> >
> > Please see our response to Weakness 4.
> >
> > ### Question 4
> >
> > Please see our response to Weakness 1 on justifications for why the present work focuses on **scientific investigation** of STAC in **controlled, python sandboxes** rather than real-world agentic deployments.

---

### Official Review · Reviewer_aDEf · 2025-10-31

**Soundness:** 2
**Presentation:** 2
**Contribution:** 2
**Rating:** 2
**Confidence:** 4

**Summary:**

This paper introduces Sequential Tool Attack Chaining (STAC), a multi-turn attack framework that exposes security vulnerabilities in LLM-based autonomous agents with tool-use capabilities. The key insight is that individual tool calls that appear benign in isolation can be chained together across multiple turns to execute harmful operations that only become evident at the final step. Through automated generation and systematic evaluation of 483 STAC cases encompassing 1,352 user-agent-environment interactions across diverse domains and 10 failure modes, the researchers demonstrate that state-of-the-art agents like GPT-4.1 are highly susceptible to these attacks, with success rates exceeding 90%.

**Strengths:**

1. The experimental design is notably thorough. The authors generated 483 distinct STAC cases encompassing 1,352 interaction sequences, covering multiple dimensions including diverse domains, task types, agent architectures, and 10 different failure modes.

2. The paper contributes by constructing a dataset of multi-turn attack scenarios specifically designed for tool-enabled agents. Unlike datasets focused on single-turn jailbreaks or standard adversarial examples, this collection captures sequential attack chains where harm emerges through composition rather than individual actions

**Weaknesses:**

My main concern is about the novelty and the threat model.

1. The paper's fundamental threat model appears flawed or insufficiently justified. In Figure 1, it seems the legitimate user is the attacker, which raises the question: if users intentionally issue these commands to their own agents, isn't the system simply following instructions as designed? For example, Turn 3 shows file deletion—but if a user wants to delete their files, the agent should comply. The paper fails to clearly establish who the adversary is, what security boundaries are being violated, or why benign users would accidentally trigger these chains. Without distinguishing between malicious injection by third parties versus intentional user actions, the high attack success rates (>90%) may simply demonstrate that agents are good at following multi-step instructions—a feature, not a vulnerability. The work lacks a clear attacker model specifying capabilities and security assumptions.

2. The core contribution appears incremental rather than novel. Multi-turn jailbreaking attacks have been extensively studied; this work primarily shifts focus from "harmful content generation" to "harmful environmental actions," which seems like a natural extension of known vulnerabilities rather than a fundamental innovation. Also, the dataset construction relies heavily on two previous works, further questioning originality. If the threat model is essentially the same as prior jailbreaking research (inducing models to perform prohibited actions), then chaining tool calls is merely a different manifestation of existing attack vectors.

3. Additionally, while the paper demonstrates that a reasoning-driven defense prompt reduces ASR by 28.8%, it would strengthen the work to explore whether this represents a robust long-term solution or if adaptive attacks could circumvent it. The paper would benefit from evaluating adaptive attacks designed specifically to break the proposed defense.

**Questions:**

I think the paper could be more impactful by exploring how STAC performs against indirect prompt injection attacks. These are cases where attackers manipulate external resources (like emails, web pages, or documents) that agents process, rather than directly controlling the prompts themselves. This type of attack seems particularly relevant for production systems since it represents a more realistic security boundary that adversaries would actually encounter.
To be clear, do you think STAC is also effective against indirect prompt injection?

---

> ### Author Response · Authors · 2025-12-03
> **Response to Reviewer aDEf (Part 1)**
>
> We thank the reviewer for their constructive feedback. We address each weakness and question raised below.
>
> ---
>
> ### Weakness 1
>
> We appreciate the reviewer’s request for a clearer articulation of the threat model. Our work does **not** assume that a legitimate user would intentionally act against their own interest. Instead, the attacker in our setting is a **malicious actor (human or agent) who has obtained access to the user-facing interface** of an agentic system. This adversary issues prompts to the agent *as the “user,”* just as in standard prompt-injection or red-teaming threat models. The question is therefore not whether an agent should obey *legitimate* user commands, but whether it can recognize when a sequence of *apparently legitimate* requests is being used to orchestrate an unintended harmful outcome.
>
> This threat model is realistic and well-aligned with contemporary security concerns. For example, Anthropic’s recent public analysis of AI-enabled espionage (https://www.anthropic.com/news/disrupting-AI-espionage) describes real-world attack patterns where adversaries intentionally **“break down their attacks into small, seemingly innocent tasks that the model will execute without being provided the full context of their malicious purpose”** through the Claude code agent. This pattern is precisely the phenomenon STAC systematizes in the agentic, tool-using setting.
>
> The security boundary being violated is therefore **the system’s ability to detect and refuse harmful high-level outcomes even when the harmful intent is distributed across multiple steps**. In many deployments, agents are expected to operate with partial autonomy (e.g., customer-support agents, personal productivity agents, workflow automation systems), and third-party access to the “user” role is both possible and documented in prior work on prompt injection, indirect prompt manipulation, cross-application attacks, and compromised integrations.
>
> Our results show that current agents lack the ability to reason about **sequence-level harmfulness**, enabling malicious actors to exploit them by decomposing harmful goals into benign-appearing tool calls. This is not simply “agents following instructions,” but a failure to enforce expected safety boundaries when malicious users disguise harmful intent over time.
>
> To avoid misunderstanding, we’ll clarify this threat model explicitly in the revision and emphasize:
>
> - The attacker is **not** the legitimate end-user but a malicious actor controlling the user channel.
> - The vulnerability concerns **sequence-level safety**, not the agent’s willingness to follow legitimate user commands.
> - The attack mirrors documented real-world adversarial behaviors where harmful intent is hidden through incremental, innocuous steps.
>
> ### Weakness 2
>
> We appreciate the reviewer’s concern and agree that harmful-content jailbreaks are well-studied. However, STAC is not a trivial extension of prior work; it addresses a *fundamentally different problem setting* with distinct challenges and safety implications. We summarize the conceptual and methodological innovations below.
>
> **1. From content harm to *state-dependent environmental harm*.**
> Prior multi-turn jailbreaks evaluate harmfulness solely at the level of **generated text**. In STAC, harmfulness emerges from the **sequence of environment state transitions** produced by tool calls. Whether an action is harmful depends on the *current environment state*, making the attack fundamentally **stateful**, not static. This represents a conceptual shift from text generation to **consequential tool-mediated action**—an area not addressed in prior jailbreak literature.
>
> **2. From single-modality prompting to *multi-modality over text, actions, and states*.**
> Traditional jailbreaks operate in a single, text-in/text-out modality. By contrast, STAC attacks unfold across:
> - natural-language prompts (input modality),
> - tool calls and parameterized actions (output modality), and
> - latent environment states (state modality).
>
> Coordinating multi-turn attacks across these modalities is nontrivial and introduces a **new class of vulnerabilities** rooted in the interaction between agent reasoning, tool affordances, and environment dynamics.
>
> **3. A substantially larger, compositional attack space.**
> Harmful text categories are limited and largely context-independent. Harmful action sequences are:
> - **compositional** across multiple tool calls,
> - **contextual** (harm emerges from the cumulative chain), and
> - **environment-dependent**, with thousands of distinct malicious outcomes possible.
>
> STAC introduces the first *systematic, automated pipeline*, including chain generation, environment verification, and adaptive execution, to tackle this much larger and more complex space. This moves beyond pattern-matching jailbreak prompts into **generalizable attack synthesis over action sequences**.

---

> > ### Author Response · Authors · 2025-12-03
> > **Response to Reviewer aDEf (Part 2)**
> >
> > **4. New empirical findings specific to agents—not LLMs.**
> > Our results show that:
> > - even models that resist text-based jailbreaks fail catastrophically on STAC,
> > - multi-turn context verification is critical for action-level vulnerabilities, and
> > - prompt-based defenses degrade over turns in an action-space setting.
> >
> > These failure modes do **not** appear in prior text-only jailbreaks and highlight a qualitatively different vulnerability class.
> >
> > **5. On the use of prior environments.**
> > SHADE-Arena and Agent-SafetyBench provide *simulation substrates*, not attack logic. Our contribution lies in:
> > - defining STAC as a new attack class,
> > - constructing the first **automated multi-turn tool-attack generator**,
> > - verifying actionable tool chains in-environment, and
> > - releasing 483 validated multi-step attack cases across 10 failure modes.
> >
> > The methodology, **not the environments**, is the core intellectual contribution.
> >
> > ---
> >
> > Together, these distinctions place STAC as more than a “manifestation” of known jailbreak vectors. It reframes the problem around **sequential tool-use vulnerabilities**, introduces a principled pipeline for generating and validating multi-step attacks, and reveals new failure patterns specific to agentic systems. We will emphasize these contributions more clearly in the revised version.
> >
> > ### Weakness 3
> >
> > We agree that assessing whether adaptive attacks can circumvent the reasoning-based defense is an important question for determining long-term robustness. However, the primary focus of this paper is on **developing and systematizing the attack methodology**, not on proposing a fully mature defense solution. The prompt-based defenses we include serve only as **baselines**, demonstrating that even relatively strong in-context defenses still struggle in the multi-turn, state-dependent setting.
> >
> > A rigorous evaluation of adaptive attacks targeting this defense, or development of stronger external defenses, would require a substantially expanded research effort. We view this as a crucial direction for **future work**, and our STAC framework provides exactly the foundation needed for constructing and evaluating such adaptive defenses.
> >
> > ---
> >
> > ### Question
> >
> > We appreciate the reviewer highlighting the relevance of indirect prompt injection for production systems. As clarified in our response to Weakness 1 re: the earlier threat-model concern, STAC is designed for a **specific and narrower threat model**: an adversary who has obtained control of the user-facing channel and can directly issue prompts to the agent. Under this model, *indirect* prompt injection is **out of scope**, and evaluating against it would require different assumptions about attacker capabilities, system architecture, and trust boundaries.
> >
> > That said, indirect prompt injection is an important attack vector for real-world deployments, and we agree it is a valuable direction for future research. While STAC was not designed with this setting in mind, its core mechanism—distributing harmful intent across benign-appearing steps—may interact with indirect injections in interesting ways. We consider a systematic exploration of this broader threat landscape to be outside the scope of the present paper but a promising extension enabled by the STAC framework.

---

### Official Review · Reviewer_QDfw · 2025-11-01

**Soundness:** 2
**Presentation:** 3
**Contribution:** 3
**Rating:** 6
**Confidence:** 3

**Summary:**

This paper introduces Sequential Tool Attack Chaining (STAC), a novel framework for studying security vulnerabilities in tool-enabled LLM agents. Unlike prior single-turn jailbreaks or prompt injections, STAC orchestrates sequences of individually benign tool calls that cumulatively lead to harmful outcomes. For example, compressing → deleting → bulk-cleaning files can collectively destroy critical data while appearing harmless in each step.

The authors design an automated pipeline that (1) generates verified tool chains via environment interaction, (2) reverse-engineers corresponding benign prompts, and (3) executes adaptive multi-turn attacks in realistic agentic settings such as SHADE-Arena and Agent-SafetyBench. The resulting STAC benchmark includes 483 validated attack trajectories spanning 10 failure modes and 1,352 user-agent-environment interactions.

Empirical results show that even frontier models (e.g., GPT-4.1, Llama 3.1, Qwen3-32B) exhibit attack success rates (ASR) above 90%, revealing a systemic vulnerability in current tool-based agent architectures. The authors also evaluate existing prompt-based defenses and propose a reasoning-based defense prompt that reduces ASR by up to 28.8%. The study argues that defending LLM agents requires reasoning over entire action sequences, not individual prompts.

**Strengths:**

The paper’s major strengths include:
1. Novel problem formulation: Identifies a new class of vulnerabilities unique to multi-turn tool-use sequences rather than individual requests.
2. Automated and reproducible pipeline: Introduces a closed-loop system for generating, validating, and testing attacks without manual curation.
3. Strong empirical evaluation: Tests across eight models, multiple environments, and failure modes, showing consistent and alarming vulnerabilities.
4. Meaningful defense proposal: The reasoning-based defense prompt is conceptually sound and shows significant mitigation without retraining.

**Weaknesses:**

While technically robust, several aspects limit the completeness of the contribution:

1. Ambiguity in novelty positioning relative to prior work.
While STAC introduces the concept of Sequential Tool Attack Chaining, the underlying behavioral bias—that models become more permissive when malicious intent is framed through code or tool interfaces—has been previously documented. The core advancement here lies in extending that phenomenon to multi-turn, executable, and agentic contexts, yet this distinction is not emphasized clearly. The paper would benefit from a sharper articulation of what conceptual or methodological gap STAC fills beyond prior tool-primed or function-call jailbreak research, making its incremental contribution explicit.

2. Limited mechanistic explanation of failure.
The paper convincingly demonstrates that sequentially benign tool calls can cumulatively yield harmful outcomes, but it provides little insight into why models fail to detect these aggregated risks. There is no analysis of whether the problem stems from reasoning myopia, context-window truncation, or loss of refusal memory across planning steps. Including introspective traces, intermediate reasoning logs, or causal attributions would elevate the contribution from empirical observation to mechanistic understanding, aligning better with ICLR’s expectation for scientific insight.

3. Overgeneralized claims of universality.
Although the results are impressive, the paper sometimes implies that STAC-style vulnerabilities generalize to all agentic deployments. Given that all experiments occur in controlled Python-based environments, this generalization may be overstated. The authors should more precisely define the boundary conditions under which STAC attacks apply, e.g., when agents have composable tools without centralized oversight or semantic validation. This would prevent overextension of the conclusions and help practitioners understand when such failures are most likely.

4. Lack of cross-model interpretive analysis.
While multiple model families are evaluated, the study largely reports aggregate metrics without examining why certain models (e.g., GPT-4.1 vs. Llama-3.1) differ in vulnerability or refusal persistence. Understanding whether these gaps stem from training data, architecture, or safety fine-tuning could yield deeper insight into agentic robustness. Analyzing a few representative trajectories across models, highlighting distinct reasoning or memory patterns, would transform the results from descriptive benchmarking into comparative scientific analysis.

**Questions:**

1. How does STAC extend or differ from prior works? Please clarify what is genuinely new in the sequential tool-chaining framework.
2. Why do models fail to detect cumulative harm across tool calls? Is this mainly due to loss of refusal memory, lack of global reasoning, or limitations in safety alignment?
3. The reasoning-based defense quickly loses effectiveness over turns. Do you have any evidence or hypothesis explaining this degradation?
4. How do you verify that a “successful” execution actually represents a harmful or policy-violating action rather than a benign result?
5. What safeguards or access policies will be in place when releasing the STAC benchmark to prevent dual-use misuse?

---

> ### Author Response · Authors · 2025-12-03
> **Response to Reviewer QDfw (Part 1)**
>
> We thank the reviewer for helpful and constructive feedback. We address each weakness and question raised below.
>
> ---
>
> ### Weakness 1
>
> We appreciate the reviewer’s request for a clearer articulation of what conceptual and methodological gaps STAC fills beyond prior tool-primed or function-call jailbreak work. We highlight the key distinctions below that contribute to the novelty of our work.
>
> **1. From text-only harm to *state-dependent* harm.**
> Prior multi-turn jailbreaks target harmful **content generation**, where harmfulness is determined solely by the text output. In contrast, STAC conceptualizes harm as **contextual over the evolving environment state**. Whether an action is harmful depends on prior tool calls and the environment’s latent state (e.g., Figure 1). This makes the attack problem fundamentally *stateful*, which is not addressed in earlier jailbreak work.
>
> **2. From uni-modal prompting to multi-modal agentic control.**
> Extending multi-turn attacks from harmful text to harmful *actions* is non-trivial. STAC operates over:
> - **text prompts** (input modality),
> - **tool calls/actions** (output modality), and
> - **environment state transitions** (latent modality).
> This introduces a multi-modal, sequential-decision structure absent from prior jailbreaks that operate in a single text modality.
>
> **3. A larger and compositional attack space.**
> Harmful text is comparatively limited and mostly universal across contexts. Harmful *action sequences*, however, expand the attack space dramatically because actions:
> - depend on environment-specific states,
> - compose across multiple steps, and
> - may only reveal harm cumulatively rather than in any single step.
> STAC specifically addresses this larger, compositional search space through verified tool chains and a synthetic multi-turn context—capabilities not present in prior jailbreak work.
>
> Taken together, these factors position STAC not as a small extension of prior findings, but as the first systematic framework that *operationalizes multi-turn, stateful, tool-mediated harm* in agentic systems. We will clarify this distinction more explicitly in the manuscript.
>
> ### Weakness 2
>
> We agree that understanding *why* models fail (e.g., reasoning myopia, loss of refusal memory, or insufficient aggregation over multi-step context) is an important scientific question, especially for developing principled defenses. However, this is out of scope for the present work.
>
> Our primary contribution is to **introduce, formalize, and systematically evaluate** a new class of multi-turn, state-dependent tool-attack trajectories. Establishing that these attacks are *feasible, general, and highly effective* across environments and agent families is already a substantial contribution. A mechanistic analysis would require a separate and significantly expanded research program involving controlled ablations, model-introspection probes, tracing architectures, or representational diagnostics—each deserving space and rigor beyond what a single paper can support.
>
> We view mechanistic investigation as a natural and valuable direction for **future work**, and our STAC benchmark provides exactly the platform needed to pursue such analysis.
>
> ### Weakness 3
>
> We appreciate the concern regarding scope and generality. To clarify, we do **not** claim that STAC vulnerabilities apply to *all* agentic deployments. In fact, we explicitly acknowledge in the paper that our evaluation is limited to controlled Python-based environments (SHADE-Arena and Agent-SafetyBench), and we frame our findings as evidence of a *class* of vulnerabilities, not a universal guarantee of failure in every deployment context.
>
> Regarding the reviewer’s suggested boundary conditions (e.g., lack of centralized oversight or semantic validation): these mechanisms represent **potential defenses**, the lack of which is not a prerequisite for our attack to succeed. Their effectiveness against STAC remains untested, and, given that STAC distributes intent across benign, state-dependent steps, it is unclear that such mechanisms would reliably catch the harmful sequence even if present. Therefore, we believe it would be premature to assert that STAC requires the absence of such controls.

---

> > ### Author Response · Authors · 2025-12-03
> > **Response to Reviewer QDfw (Part 2)**
> >
> > ### Weakness 4
> >
> > We agree that deeper interpretive comparisons across model families would be scientifically valuable. In our case, however, the empirical landscape leaves limited room for such analysis: **STAC achieves very high ASR (>90%) on nearly all models**, and the differences between model families are correspondingly small. The main exception is Magistral, where lower ASR stems not from stronger safety behavior but from **insufficient capability to complete certain multi-step tool operations**, as confirmed by our qualitative inspection of trajectories.
> >
> > Because the attack succeeds so consistently across architectures, training regimes, and safety pipelines, there is relatively little variation to analyze in terms of refusal persistence or reasoning divergence. Our primary aim in this work is to demonstrate the *existence* and *generality* of the vulnerability class. We agree that more in-depth, cross-model mechanistic comparison is an important direction for future work, and our benchmark provides a foundation for such analyses.
> >
> > ---
> >
> > ### Question 1
> >
> > Please refer to our response to Weakness 1, which further clarifies the novelty of our framework compared to prior multi-turn attack work.
> >
> > ### Question 2
> >
> > Please refer to our response to Weakness 2.
> >
> > ### Question 3
> >
> > We appreciate the question. We do not yet have concrete empirical evidence pinpointing the cause of the degradation, and a full investigation is beyond the scope of this paper. Our preliminary hypothesis is that the decline stems from the **prompt-based nature** of the defense (e.g., through context manipulation).
> >
> > If the same reasoning procedure were implemented as an **external guardrail**, rather than as an in-context prompt, it may maintain more stable effectiveness across turns. We view systematic evaluation of guardrail architectures and inference-time safety modules as important future work.
> >
> > ### Question 4
> >
> > To ensure that a “successful” execution corresponds to an actual harmful or policy-violating action, we rely on a **Judge** model (GPT-4.1) that evaluates both the agent’s tool call and the resulting environment change. GPT-4.1 was chosen for its strong reasoning and safety capabilities, which provide relatively consistent and high-quality judgments across cases.
> >
> > To validate this automated evaluation, we **manually inspected a subset of trajectories** across different environments and failure modes. In these cases, the Judge’s assessments aligned with human judgments overall, giving us confidence that the automated scoring reliably distinguishes harmful outcomes from benign ones.
> >
> > ### Question 5
> >
> > This is an important concern. We will release the STAC benchmark under a **non-commercial, research-only license**, restricting use to academic and safety-oriented work. In addition, all STAC trajectories are grounded in **controlled Python simulation environments** rather than real systems, so the data cannot be directly applied to actual deployments or live infrastructures. The environments, tools, and APIs in STAC are synthetic and intentionally simplified to study vulnerability patterns rather than to replicate production interfaces.

---

### Official Review · Reviewer_yhkJ · 2025-11-01

**Soundness:** 3
**Presentation:** 3
**Contribution:** 3
**Rating:** 6
**Confidence:** 3

**Summary:**

The paper introduces STAC (Sequential Tool Attack Chaining), an automated, closed-loop framework that chains together harmless tool calls, which collectively enable harmful operations at the final execution step. Experiments show that on SHADE-Arena and Agent-SafetyBench, STAC achieves very high ASR. A reasoning-based defense prompt reduces ASR by up to 28.8% on the first execution turn, but protection degrades over turns.

**Strengths:**

1. The framework is clearly formalized. Generator/Verifier/Prompt Writer/Planner/Judge Roles are explicit and reproducible。

2. The proposed framework performs effectively across 4 complex SHADE-Arena environments (banking, travel, workspace, spam filtering) plus 62 Agent-SafetyBench environments.

3. The threat is operational: not just generating unsafe text, but orchestrating tool use to cause environmental harm.

4. In addition to proposing the attack framework, the paper also includes a defense analysis.

**Weaknesses:**

1. The attack baselines are limited. There are many multi-turn jailbreakers and agent-focused attacks that are not included. The paper adapts only X-Teaming. The comparison with X-Teaming is also potentially unfair. The adapted X-Teaming setup allows the agent to take only one tool-call turn per user turn, “due to system design limitations,” which likely handicaps it relative to STAC’s adaptive multi-turn execution. Moreover, hiding intentions in a multi-turn attack to improve ASR is not entirely new. While the paper focuses on tool-using settings, it is crucial to demonstrate the method's advantage over other baselines across various domains.

2. Only prompt-based defenses are evaluated. It seems that defenses such as tool-policy or allow/deny lists can be effective against at least some of the attacks (e.g., the example in Figure 3).

3. GPT-4.1 is used as Generator, Verifier, Planner, and Judge, which might raise concerns about single-model dependence in the attack pipeline and evaluation

**Questions:**

1. How does ASR evolve beyond 3 user turns? Is it possible to report a turn-by-turn curve until a fixed horizon (e.g., 5–10 turns) to illustrate persistence effects?

2. What happens when tools themselves enforce policies (e.g., “delete_file” requires a capability token or asks for user confirmation if the path is critical)?

**Details Of Ethics Concerns:**

The paper proposes an attack method for LLM agents.

---

> ### Author Response · Authors · 2025-12-03
> **Response to Reviewer yhkJ (Part 1)**
>
> We thank the reviewer for their thoughtful and constructive feedback. We address each point in detail below.
>
> ---
>
> ### Weakness 1
>
> **1. On available multi-turn baselines.**
> At the time of our study, no multi-turn jailbreak targeting **agent tool use** baselines existed. Prior multi-turn jailbreaks target **harmful content generation**, not **harmful tool-use actions**, and applying them directly would evaluate a different task. Environment-specific “agentic attacks” (e.g., web, code agents) also do not generalize to the diverse tool-enabled environments we evaluate, making them unsuitable as baselines.
>
> **2. Why we used X-Teaming.**
> X-Teaming was the strongest general multi-turn jailbreak available and the closest methodological match. We also included a single-turn baseline to isolate the effect of multi-turn context.
>
> **3. On adapting X-Teaming.**
> Our adaptation already extends X-Teaming to handle tool feedback. Allowing multiple agent turns per user turn is *incompatible* with the core **textgrad** update, which requires a single, direct model output to optimize the user prompt. Removing textgrad would significantly weaken X-Teaming and result in an unfair comparison.
>
> **4. Fairness of comparison.**
> We therefore benchmark X-Teaming in its strongest feasible form for the agentic setting—preserving textgrad while integrating tool feedback. Even under this favorable adaptation, X-Teaming is substantially less effective than STAC, highlighting that text-only jailbreak methods do not transfer reliably to tool-enabled agents.
>
> **5. On “hidden intent” being known.**
> While intent diffusion is familiar in text jailbreaks, STAC is distinct in (i) chaining *verified* benign tool calls, (ii) ensuring full in-environment executability, and (iii) targeting harmful *actions* rather than harmful text. This exposes a security gap unique to agentic systems.
>
> ### Weakness 2
>
> We appreciate the reviewer’s point regarding the limited scope of defenses. Our primary focus in this work is to (i) demonstrate the feasibility and effectiveness of multi-turn tool-attack chaining on agents, and (ii) develop a scalable, automated pipeline for synthesizing such attacks across diverse environments. Prompt-based defenses were included as *initial baselines* because they are lightweight, versatile, and agnostic to models, environments, and tasks.
>
> We fully agree that additional defense classes, such as tool-policy constraints, allow/deny lists, or tool-level access control, may provide stronger protection against certain attack patterns (including the example in Figure 3). A comprehensive evaluation of these mechanisms is outside the scope of this paper’s attack-centric contribution, and we explicitly acknowledge this as a limitation. We view systematic analysis and design of such defenses as important future work, and our benchmark provides the foundation on which those defense methods can be rigorously tested.
>
> ### Weakness 3
>
> We acknowledge the concern about using GPT-4.1 across all roles. Our design choice is intentional for two reasons:
>
> **(1) Consistency within the attack pipeline.**
> The Generator, Verifier, and Planner collectively form the *attacker scaffold*, each specializing in different subtasks of the same attack process. Using the same underlying model ensures consistent reasoning, avoids cross-model disagreement, and reduces variance introduced purely by model heterogeneity. This leads to a cleaner, more consistent attack generation.
>
> **(2) Consistency between generation and evaluation.**
> Using the same model as the Judge ensures that the attacker and the evaluator share a consistent understanding of what constitutes “harmful actions.” Heterogeneous Judge models might disagree on what constitutes an attack with the generator, which would inject noise or bias into the measured attack success rates.
>
> We agree that exploring diverse model configurations is a valuable direction, but our aim in this paper is to present a principled and controllable evaluation pipeline. Future work can incorporate heterogeneous models to examine robustness under cross-model variation to extend the scope of the framework.

---

> > ### Author Response · Authors · 2025-12-03
> > **Response to Reviewer yhkJ (Part 2)**
> >
> > ### Question 1
> >
> > In preliminary experiments, we observed that ASR **plateaus after the third attack-execution turn is reached**. By Turn 3, ASR already exceeds **90% for nearly all models**, leaving little room for additional improvement. Because the attack is already highly effective at this point, extending to 5–10 turns provides minimal additional signal while substantially increasing cost. Given this saturation behavior and practical resource constraints, we focus our evaluation on the first three attack execution turns.
> >
> > ### Question 2
> >
> > This is an excellent question. The environments we use (SHADE-Arena and Agent-SafetyBench) do not currently support tool-level policies such as capability tokens, critical-path confirmations, or dynamic access control. These frameworks represented the state of the art environments at the time of our study, but they intentionally model tools as callable functions without built-in security policies.
> >
> > Evaluating STAC under richer tool-governance mechanisms is an important direction for future work, and we would be very interested in testing our framework on environments that incorporate capability-based restrictions or tool-specific safety protocols once they become available.

---

### Public Comment · ~Pengyu_Zhu1 · 2025-11-18

Hi, Authors, I appreciate your insightful work. However, I noticed that DemonAgent, a closely related study, appears to be overlooked in your references. Incorporating a discussion on the distinctions between your proposed method and DemonAgent would significantly strengthen your paper and provide a more comprehensive context for readers.

---

> ### Author Response · Authors · 2025-12-03
> **Response to Public Comment by Pengyu Zhu**
>
> Dear Pengyu,
>
> Thank you for your suggestion. We apologize for overlooking this related work and will reference it in the next version of our paper.
>
> Best,
> Authors

---

### Note · Authors · 2026-01-06

I have read and agree with the venue's withdrawal policy on behalf of myself and my co-authors.